# Using Machine Learning to Predict Bacteremia in Febrile Children Presented to the Emergency Department

**DOI:** 10.3390/diagnostics10050307

**Published:** 2020-05-15

**Authors:** Chih-Min Tsai, Chun-Hung Richard Lin, Huan Zhang, I-Min Chiu, Chi-Yung Cheng, Hong-Ren Yu, Ying-Hsien Huang

**Affiliations:** 1Department of Pediatrics, Kaohsiung Chang Gung Memorial Hospital and Chang Gung University College of Medicine, Kaohsiung 833401, Taiwan; tcmnor@cgmh.org.tw (C.-M.T.); yuu2002@cgmh.org.tw (H.-R.Y.); 2Department of Computer Science and Engineering, National Sun Yat-sen University, Kaohsiung 804201, Taiwan; m063040001@g-mail.nsysu.edu.tw (H.Z.); ray1985@cgmh.org.tw (I.-M.C.); qzsecawsxd@cgmh.org.tw (C.-Y.C.); 3Department of Emergency Medicine, Kaohsiung Chang Gung Memorial Hospital and Chang Gung University College of Medicine, Kaohsiung 833401, Taiwan

**Keywords:** machine learning, predict, bacteremia, children, emergency department

## Abstract

Blood culture is frequently used to detect bacteremia in febrile children. However, a high rate of negative or false-positive blood culture results is common at the pediatric emergency department (PED). The aim of this study was to use machine learning to build a model that could predict bacteremia in febrile children. We conducted a retrospective case-control study of febrile children who presented to the PED from 2008 to 2015. We adopted machine learning methods and cost-sensitive learning to establish a predictive model of bacteremia. We enrolled 16,967 febrile children with blood culture tests during the eight-year study period. Only 146 febrile children had true bacteremia, and more than 99% of febrile children had a contaminant or negative blood culture result. The maximum area under the curve of logistic regression and support vector machines to predict bacteremia were 0.768 and 0.832, respectively. Using the predictive model, we can categorize febrile children by risk value into five classes. Class 5 had the highest probability of having bacteremia, while class 1 had no risk. Obtaining blood cultures in febrile children at the PED rarely identifies a causative pathogen. Prediction models can help physicians determine whether patients have bacteremia and may reduce unnecessary expenses.

## 1. Introduction

Fever is one of the most frequent reasons for visits to the Pediatric Emergency Department (PED) [1,2], and estimates say that up to 10% to 25% of cases with febrile illness had bacterial infection [3,4]. Determining the appropriate method for evaluating febrile children remains a challenge, especially due to fears regarding occult bacteremia in febrile children who appear well without an obvious infection focus [5]. Bacteremia is a severe bacterial infection that can be detected by a blood culture, one of the most frequently ordered microbiological tests in the PED [6]. Blood cultures remain the gold standard test for detecting patients with bacteremia. Isolating the organism from the blood can confirm a diagnosis, which helps physicians identify the cause of the infection and then administer the appropriate antimicrobial agents. Upon receiving a blood culture result, physicians must decide whether the organism represents a clinically significant infection [6,7,8]. However, seeing a high rate of negative, false-positive, or contaminated blood cultures is quite common in children visiting the PED [9]. All of the above conditions result in the unnecessary use of healthcare resources and costs, including additional invasive testing, the inappropriate use of antibiotics, prolonged hospital admission, and parental anxiety [10,11,12,13].

Over the past few decades, several algorithms have been developed to identify children at a higher risk of severe bacterial illness [3,4,14]. In addition to clinical findings, some studies have suggested that laboratory findings, such as white blood cell count (WBC), absolute neutrophil count (ANC), C-reactive protein (CRP), and procalcitonin (PCT), may be useful in helping physicians recognize children with severe bacterial infection [4,14,15,16,17,18]. Although these parameters can help pediatric clinicians identify febrile children at high risk of severe bacterial infection, are they capable of predicting bacteremia in febrile children? Many studies also debate the role of appropriate blood cultures in the PED. Obtaining blood cultures is only recommended for children with extensive infections or immune-compromised patients or for those with moderately or severely ill children, according to published guidelines and studies [19,20].

In this study, we tried to build machine learning models to predict bacteremia in children with fever who visit the PED. Our findings can be clinically important as they may help physicians in the PED either order the appropriate blood culture or manage treatment depending on whether bacteremia is predicted.

## 2. Materials and Methods 

### 2.1. Study Population

Patients younger than 18 years of age who presented to the PED of the Kaohsiung Chang Gung Memorial Hospital in Taiwan with fever during the period of January 2008 through December 2015 were evaluated. Febrile children from whom a blood test and blood culture were obtained formed our retrospective cohort. In this cohort, each febrile child with true bacteremia was randomly matched with 10 febrile children without bacteremia according to gender and age in order to form a case control study. All the blood cultures were collected by nurses in accordance with our institution’s standard procedures. Obtaining two sets of blood cultures was quite difficult in pediatric patients, so one set of blood culture for children with fever was generally practiced in our PED. This study was approved by the Institutional Review Board of Chang Gung Medical Foundation. 

### 2.2. Blood Culture Criteria

The following organisms that were isolated from the blood sample represent a true pathogen: *Staphylococcus aureus*, *Streptococcus pneumoniae*, *Salmonella enterica*, group A streptococci, *Pseudomonas aeruginosa*, *Haemophilus influenzae*, *Escherichia coli* (*E. coli*), and Candida species, among others [21,22,23,24]. Certain organisms isolated from blood samples have been found to represent contamination. These pathogens include coagulase-negative staphylococci, *Staphylococcus epidermidis*, *Corynebacterium* spp., Gram-positive Bacillus, *Micrococcus*, etc. [21,22,23]. In the case of any doubt related to the potential pathogenicity of one of the isolated species, the research coordinator reviewed the case to determine whether the corresponding blood culture was an actual infection. 

### 2.3. Statistical Analysis

Continuous variables were expressed as mean ± standard deviation, and categorical variables were reported as percentages. To compare clinical characteristics between children with and without bacteremia, Student *t* tests and Fisher’s exact test or *x*^2^ test were used for continuous variables and categorical variables, respectively. Univariate and multivariate binary logistic regression analyses were used to identify the significant risk factors. A *P*-value less than 0.05 was considered statistically significant. IBM SPSS statistical software for Windows, version 22.0 (Chicago, IL, USA) was used for statistical analyses. 

### 2.4. Experimental Methodology

Age, gender, and laboratory values obtained from medical records were evaluated as predictive variables by the models. For every laboratory value variable, only values obtained simultaneously with the blood culture sample were used. In addition to gender and age, 17 laboratory values were included in our study population. The normal reference of these values may differ in different ages or genders, so case control study was used to eliminate these confounding factors. Due to the hugely different case numbers in the bacteremia group and non-bacteremia group, each case with bacteremia was randomly matched by age and gender with ten cases without bacteremia. To avoid sample selection bias, the matching procedure was repeated 100 times for the characteristic variable analysis. The features selected for machine learning were determined according to the 100 times of characteristic variable analysis. The selected features were further used to establish the predictive model. 

Python language, R language, and machine learning methods—logistic regression (LR) and support vector machines (SVM) were used to establish a predictive model for bacteremia. Cost-sensitive learning was applied to make a trade-off between false negative predictions and cost reduction to increase the usability of the predictive model. In the dataset, 97.5% of the cases were used for training, and 2.5% were used as testing data. A risk value of each case was calculated using the SVM or LR predictive model. Since the dataset is generally imbalanced, cost-sensitive learning is used for imbalanced classification. The costs of prediction errors (and potentially other costs) are considered when training a machine learning model. We set bacteremia as positive and non-bacteremia as negative. The amounts of positive and negative cases were quite different. Considering the confusion matrix of inference, the costs of true positive and true negative were set to zero. That is, those cases for which we can correctly predict the result had no cost. In this study, the false negative was more important, which means it will cost more when we predict bacteremia as non-bacteremia. Therefore, we must pay more attention to false negatives. If we assign the cost of false positive as 1, then we must carry out experiments to dynamically adjust the cost of a false negative (*an important parameter that can significantly affect the performance of prediction*) and find the optimal one. The optimal value is in the range of 7~13. The *risk* is defined as the cost of negative (the cost of predicting a non-bacteremia result) minus the cost of a positive (the cost of predicting a bacteremia result) (i.e., *risk = negative cost – positive cost*). If cost > 0, predict bacteremia, otherwise, predict non-bacteremia. However, this risk-based binary prediction (i.e., cost > 0 and cost < 0) does not have good performance. Instead of predicting by plus or minus of risk value, we partition the range of risk value into a couple of segments, such as the quartile, and we make a prediction for each segment; therefore, the performance can be obviously improved.

## 3. Results

### 3.1. Patient Characteristics

Among a total of 266,679 children that visited the PED during the 8-year period, 16,967 febrile children with blood culture tests were enrolled for further analysis. The mean age of the total study population was 5.16 ± 3.97 years old. Of these 16,967 children, 55.3% (*n* = 9388) were male. We observed three kinds of blood culture results: bacteremia indicating true infection, contamination indicating a false-positive result, and negative culture (no growth of wither aerobic or anaerobic pathogens). Patients were categorized into two groups according to the blood culture results: the bacteremia group and the non-bacteremia group (including contamination and negative results). In this study cohort, 146 (0.86%) febrile children had true bacteremia, 405 (2.39%) children had a contaminant result, and 16,416 (96.75%) children had a negative result. The gender, age, and laboratory tests were obtained for further analysis and compared according to the blood culture results, as shown in Table 1. Age, percentage of neutrophil and band, ANC, Hb, platelet, and CRP were statistically different between bacteremia and non-bacteremia encounters. The three most common pathogens in the bacteremia group were *Salmonella entericae* (28/146, 19.18%), *Escherichia coli* (22/146, 15.07%), and *Streptococcus pneumoniae* (14/146, 9.59%). The three most common pathogens isolated from blood culture and considered contamination were *Staphylococcus epidermidis* (137/405, 33.83%), coagulase-negative staphylococcus (129/405, 31.85%), and *Micrococcus* (45/405, 11.11%). The age distribution of the bacteremia group is shown in Figure 1. More than half (55.7%) of the febrile children with bacteremia were under the age of 3 years old.

### 3.2. Feature Selection and Risk Classification

After repeating the characteristic variable analysis 100 times, WBC, MCV, MCH, MCHC, Monocyte, Eosinophil, CRP, Band percentage, Segment + Band percentage, and ANC were all positively or all negatively correlated with bacteremia (Table 2) and their odds ratio are shown in Table 3. These 10 features were selected to establish the predictive model. Among these 10 features, significant risk factors were also picked up for multivariate binary logistic regression analysis and their coefficients are shown in Table 4. We used the cost-sensitive approach in machine learning to tackle the imbalance dataset which we have collected in this study. The curves of recall (sensitivity), true-negative rate (specificity) and AUC (area under the ROC curve) for different cost we applied are shown in Figure 2 (each point represents the average result of 100 times analyses). These three performance indexes were considered in our study. Thus, we can choose the best cost value to balance these performance indexes. The maximum values of different performance indexes by LR are shown in Table 5. With cost-sensitive learning, the maximum areas under the curve (AUC) of LR and SVM to predict bacteremia were 0.768 and 0.832, respectively. 

Considering the bacteremia samples in the dataset, we calculated a risk value of each sample and their quartile ranges are shown in Figure 3. We did the same for the risk value of non-bacteremia samples and showed its quartile range. According to the range of the minimum risk value of bacteremia −0.640) and maximum risk value of non-bacteremia (0.644), it is divided into three categories. Further analyzing the quartile range, bacteremia data are located between [−0.014, 0.15], while the non-bacteremia data are located between [−0.061, 0.132]. Using the 1st quartile of bacteremia and 3rd quartile of non-bacteremia, we can divide the range of risk value to five blocks (as shown in Figure 4). With these five classes, we can predict class 1 as being non-bacteremia, class 2 as low risk of bacteremia, class 3 as medium risk, class 4 as high risk, and class 5 as being bacteremia.

### 3.3. Subgroup Study 

Most febrile children with bacteremia were those under the age of 3 years old. Therefore, we conducted a subgroup study that included only these younger children. Among this subgroup, 1.58% (*n* = 81) febrile children had true bacteremia, 4.7% (*n* = 238) children had a contaminant result, and 93.8% (*n* = 4795) children had a negative result. Gender, age, and laboratory tests were obtained for further analysis and compared according to the blood culture results, as shown in Table 6. Age, percentage of band and eosinophil, hemoglobin and CRP differed statistically between bacteremia and non-bacteremia encounters in the subgroup study. Of these young children, the most common isolated pathogens were *Salmonella entericae* (26/81, 32.10%), *Escherichia coli* (19/81, 23.46%), and Group B Streptococcus (6/81, 7.41%). After repeating the characteristic variable analysis 100 times, the significant factors and their ORs are shown in Appendix A. The multivariate binary logistic regression coefficients of significant risk factors are shown in Appendix A.

The AUC ranged between 0.616 to 0.750 for predicting bacteremia in children under the age of 3 years old in the model we developed.

## 4. Discussion

The main findings of this study were as follows: (1) the bacteremia rate in febrile children that presented to the PED was low, (2) CRP was significantly higher and hemoglobin was significantly lower in children with bacteremia, (3) younger children (<3 years of age) with fever are more likely to have bacteremia than older children, and (4) machine learning can help us classify the risk of bacteremia in febrile children.

As many as 3–10% of well-appearing children under the age of 3 years old with fever without a source were found to have an occult bacteremia in the prevaccine era. Due to the concern of bacteremia becoming an invasive illness, many practitioners recommended routine blood tests, including blood culture, and then antibiotic therapy based on WBC results, as part of the management strategy for these children [3,14]. However, since the introduction of the *Haemophilus influenzae* type b (Hib) vaccine in the late 1980s and the pneumococcal conjugate vaccine (PCV) in the 2000s, a dramatic decline in bacteremia was observed as low as 0.25% to 1.43% in children [25,26,27,28]. Irwin et al. also reported an annual reduction of 10.6% in vaccine-preventable bacteremia and found that PCV was associated with a 49% reduction in pneumococcal bacteremia between 2001 and 2011 [29]. The Hib vaccine and PCV were first introduced to Taiwan in 1996 and 2005 respectively. In our current study, the overall bacteremia rate was about 0.86% in all febrile children and about 1.58% in younger febrile children (less than 3 years) that presented to the PED. Furthermore, only <0.1% (*n* = 14) febrile children were identified to have bacteremia with *Streptococcus pneumoniae*, and none of the blood culture results yielded Hib. This result is in agreement with the low bacteremia rate in the post-Hib vaccine and post-pneumococcal vaccine eras. Although adequate aseptic techniques can substantially reduce the risk of contaminating blood culture specimens, contamination rates of 2% to 3% are considered acceptable [30]. The overall contamination rate was 2.39% in our study, and the isolation of contaminant organisms from a blood culture has a significant negative impact on patient management, including misdiagnosis, unnecessary antibiotics, performance of additional and unnecessary diagnostic tests, additional costs, and prolonged hospital stays [11,12,13]. A low positive blood culture result with a high rate of contaminant results has made physicians doubt the usefulness of blood cultures in children with fever that visit the PED. How to reduce unnecessary blood cultures will become an important issue for healthcare systems in the postvaccine era.

CRP is an acute-phase reactant protein synthesized by the liver in response to elevated cytokine levels and has been studied as a sensitive marker of bacterial infection [31,32]. Many studies have proposed that high CRP concentration may be associated with severe bacterial infection in febrile infants and children [15,18,33,34,35]. In both the complete study population and the younger age subgroup (<3 years) in our study, elevated CRP concentration was significantly higher in patients with bacteremia. Our results support the finding of high CRP levels in children with bacterial infection. Using CRP to properly manage children with fever may help identify true bacteremia and reduce unnecessary antibiotic therapy. Some studies have used CRP together with other parameters to predict children with severe bacterial infection. In two recent studies, CRP with extreme leukocytosis was proposed to be useful in predicting severe bacterial infection in children [33,34]. Buendia et al. showed that Rochester criteria plus CRP testing was the most cost-effective strategy for detecting serious bacterial infections in children one to three months old with fever without a source [36]. However, the Rochester criteria is especially applied to young infants, not older infants and children. To the best of our knowledge, no study used more than two clinical parameters together with CRP to predict bacteremia in febrile children. We proposed a useful model for predicting bacteremia in febrile children, not only those with CRP but also other common laboratory parameters in the PED setting. 

Anemia due to disease is often seen in various inflammatory states, including acute or chronic infections, autoimmune problems, chronic kidney disease and inflammation, and certain cancers [37]. Anemia has commonly been associated with infections that are typically seen in a pediatric primary care setting [38]. In 2009, Ballin et al. also reported that bacteremia and pyelonephritis are accompanied by a significant drop in hemoglobin levels without evidence of hemolytic anemia [39]. When infection occurred, the inflammatory cytokine could induce hepcidin production in the liver, increase macrophage activation and red blood cell (RBC) destruction, and suppress erythropoiesis. Therefore, inflammation-related anemia may result from hepcidin-induced hypoferremia combined with the cytokine-mediated suppression of erythropoiesis and decreased lifespan of erythrocytes [40]. This phenomenon can explain the findings of lower Hb in children with bacteremia in our cohort.

Significant differences in the percentage of neutrophil, ANC, platelet and eosinophils were also observed in our study. However, most reference intervals of pediatric hematology analytes are age-dependent, especially WBC and its differential count [41,42]. The changes in either absolute count or percentage of neutrophils are dynamic, particularly in the young infants and during the first years of life. The mean percentage of neutrophils may as low as 31–33% and the mean count of neutrophils also achieves its nadir at an age of 6 months to 2 years old [42]. Both lower percentage of neutrophils and ANC in children with bacteremia may due to the younger age (81 of 146 cases are younger than 3 years of age) in our study population. Reactive thrombocytosis has diverse etiologies, including inflammatory, neoplastic and infectious diseases [43]. In most patient series, acute infections represent the most common cause of reactive thrombocytosis [44,45]. In addition to CRP induction, interleukin-6 also plays a pivotal role in thrombocytosis of inflammation [46]. In children with bacteremia, the inflammation-associated cytokines produced primarily by WBC at inflammatory sites may further cause the elevation of CRP level and induce reactive thrombocytosis. This can explain the finding of higher platelet count in our bacteremia group. The pathophysiology of eosinopenia is related to the migration of eosinophils to the inflammatory site, presumably as a result of chemotactic substances secreted during the acute phase of inflammation [47]. A decreased number of circulating eosinophils is regarded as a consequence of acute bacterial infection and several studies have used eosinophil count as an indicator of bacteremia [48,49,50,51]. Our finding supports the view of low eosinophil count in patients with bacteremia. 

Zeretzke et al. reported that the children most at risk for occult bacteremia are those younger than 36 months of age with a fever of 39 °C or higher [52] due to the high probability of developing serious bacterial infections, such as meningitis, sepsis, pneumonia, septic arthritis, osteomyelitis, and pyelonephritis [5,14,53]. Therefore, obtaining blood cultures for febrile children with a young age is reasonable. However, febrile children with bacteremia are mostly seen in children under the age of 3 years old with a bacteremia rate of 1.58% versus 0.55% in children more than 3 years old in our study. In other words, these older febrile children with lower probability of bacteremia may have more unnecessary blood cultures, which may waste medical resources. How to reduce the frequency of blood culture in febrile children without misdiagnosis of bacteremia becomes an important issue.

The quality and cost of the healthcare being provided has become an increasing issue worldwide. This concern has led to a focus on how we can achieve equal or better quality outcomes with fewer health resources or less money. Segal et al. described contaminant blood cultures in 85 children that added more than $78,000 in unnecessary charges [13]. A recent study also demonstrated a yearly savings of ∼$250,000 in hospital charges when the blood culture contamination rate was reduced from 3.9% to 1.6% [11]. Some guidelines for inpatient community-acquired pneumonia (CAP) management recommend considering blood culture testing for inpatients with moderate to severe bacterial pneumonia [19,20]. However, obtaining blood cultures in children hospitalized with CAP rarely identifies a causative pathogen, which makes blood cultures less useful [54,55]. The high rates of negative culture results can also represent overuse. In our current study, we have also found a high negative culture rate and a low bacteremia rate, which indicates an overuse of blood culture and a waste of healthcare resource in febrile children. Therefore, how to reduce the over-use of blood culture without missing patients with dangerous bacteremia is important. With the “cost-sensitive learning” model that we proposed with machine learning, we can identify those febrile children with no risk of bacteremia (class 1 risk value) and avoid unnecessary blood cultures to save healthcare resources. 

Blood culture remains the gold standard to diagnose bacteremia, but it is a time-consuming diagnostic tool. After the blood sample being collected, it may take couple days to have the initially result of Gram stain, such as Gram-positive cocci, Gram-negative bacilli, etc. Physicians may be informed the final result of blood culture further few days later. Those features used in our prediction model are laboratory data which can be available within one hour after blood sample being collected. In clinical practice, we can implement a decision-making application program running over personal computer as a decision support tool. Those variables of laboratory findings are input data, and this tool can give a report to illustrate a risk probability of bacteremia for clinical reference. In addition, the report will also come out with the distribution of each variable of patients in the database in a way of data visualization for comparison. Therefore, our prediction model can be a part of clinical decision support system to help physicians determine whether patients have risk of bacteremia and thereafter arrange adequate medical treatment.

The results of this study should be interpreted with respect to certain limitations. First, procalcitonin (PCT), a useful biomarker proposed for predicting bacterial infection [56], was not commonly used in our hospital during the study periods. Therefore, PCT use was noted as a variable in our models. Second, those febrile patients who visited out-patient departments and were hospitalized were not included. These patients may also have bacteremia. Third, the models that we used mostly relied on laboratory tests, and the information contained within medical notes were not used. Fourth, some information recorded in medical notes, such as clinical symptoms, location of infection, vital sign (heart rate, respiratory rate, blood pressure, and oxygen saturation), respiratory pattern, and general appearance of the patient, etc. are important to help physician to determine the severity of a febrile patient. However, these data were not available in our database to improve our prediction model. These might have limited our model’s performance. Natural language processing techniques to get bacteremia-relevant information from unstructured medical notes are expected to improve the predictive models.

## 5. Conclusions

Obtaining blood cultures in febrile children at the PED are definite diagnosis of bacteremia but they rarely identify a causative pathogen. Moreover, overuse or waiting the result of blood culture has been described in relation to a financial burden to healthcare system. Our machine learning prediction model can be a part of clinical decision support system and help physicians determine whether patients have risk of bacteremia and may reduce unnecessary expenses.

## Figures and Tables

**Figure 1 diagnostics-10-00307-f001:**
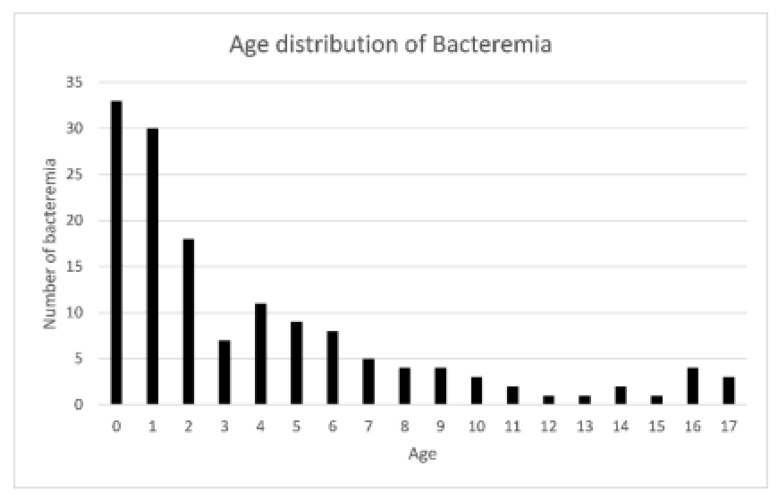
The distribution of bacteremia at different ages of the study population.

**Figure 2 diagnostics-10-00307-f002:**
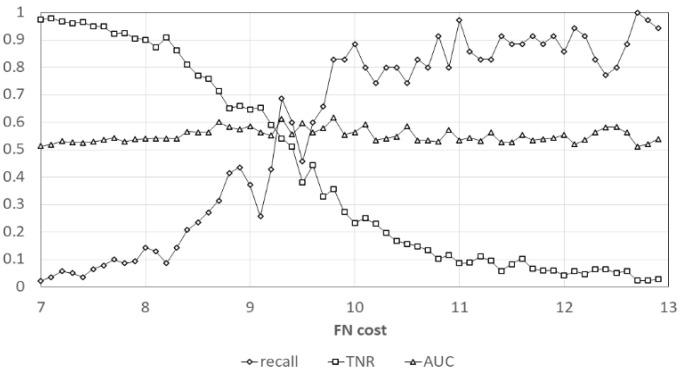
The relation of FN cost and recall (sensitivity), TNR (specificity), AUC in our predictive model. FN, false negative; TNR, true negative rate; AUC, area under ROC curve.

**Figure 3 diagnostics-10-00307-f003:**
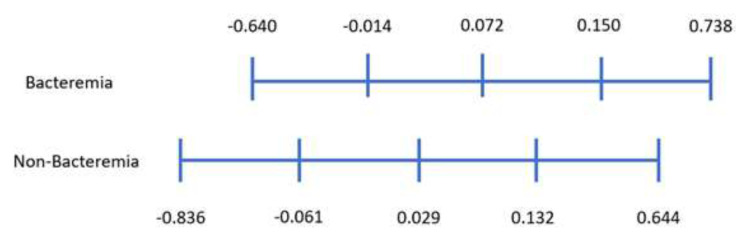
The range of risk value and its quartile range.

**Figure 4 diagnostics-10-00307-f004:**
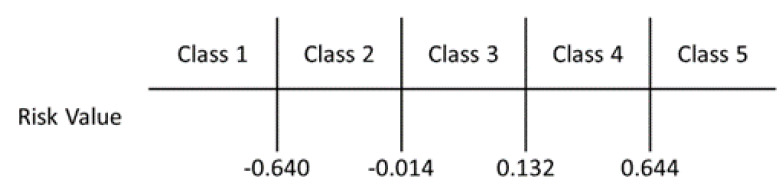
The probability of being bacteremia classified by risk value.

**Table 1 diagnostics-10-00307-t001:** Distribution of several variables in febrile children with or without bacteremia.

Variable	Bacteremia	Non-Bacteremia	*p*-Value
Demographics	N = 146	N = 16,821	
Age	3.85 ± 4.50	5.17 ± 3.97	<0.001
Gender = Male	80 (54.79%)	9308 (55.34%)	0.896
Gender = Female	66 (45.21%)	7513 (44.66%)	
Laboratory Values			
WBC (10^3^/μL)	12.14 ± 6.92	11.00 ± 7.33	0.062
Neutrophil (%)	56.35 ± 24.04	62.06 ± 19.67	0.005
Lymphocyte (%)	28.93 ± 19.18	26.64 ± 16.37	0.152
Band (%)	0.76 ± 1.86	0.34 ± 1.54	0.007
Monocyte (%)	7.54 ± 5.38	7.80 ± 3.98	0.561
Eosinophil (%)	0.69 ± 1.28	0.93 ± 1.57	0.067
Basophil (%)	0.22 ± 0.33	0.24 ± 0.36	0.501
ANC (10^3^/μL)	8.57 ± 3.60	9.36 ± 2.95	0.009
Hemoglobin (g/dL)	11.68 ± 1.63	12.27 ± 1.35	<0.001
MCV (fL)	80.51 ± 7.50	79.76 ± 6.66	0.179
MCH (pg)	27.19 ± 3.02	26.95 ± 2.55	0.339
MCHC (g/L)	33.71 ± 1.38	33.75 ± 1.02	0.646
Platelet (10^3^/μL)	279.28 ± 119.06	255.50 ± 93.04	0.017
AST (U/L)	42.88 ± 36.44	37.45 ± 39.94	0.101
ALT (U/L)	27.75 ± 28.15	22.35 ± 34.04	0.056
CRP (mg/L)	53.59 ± 68.24	36.46 ± 52.04	0.003

WBC, white blood cell; ANC, absolute neutrophil count; MCV, mean corpuscular volume; MCH, mean corpuscular hemoglobin; MCHC, mean corpuscular hemoglobin concentration; ALT, alanine transaminase; AST, aspartate transaminase; CRP, C-reactive protein.

**Table 2 diagnostics-10-00307-t002:** The maximum value, minimum value, mean value, and standard deviation of logistic regression coefficients of characteristic variables.

Variable	Minimum–Maximum			(Mean ± SD)				
AGE	−0.0092	–	0.0369	(	0.0155	±	0.0084	)
WBC	−0.1275	–	−0.0323	(	−0.0822	±	0.0176	)
CRP	0.0020	–	0.0059	(	0.0039	±	0.0008	)
Hemoglobin	−0.4393	–	0.3613	(	0.0119	±	0.1562	)
MCV	−0.7722	–	−0.2038	(	−0.5010	±	0.1216	)
MCH	0.4870	–	2.1123	(	1.4471	±	0.3296	)
MCHC	−1.7403	–	−0.3591	(	−1.0818	±	0.2826	)
Platelet	−0.0002	–	0.0012	(	0.0004	±	0.0003	)
Lymphocyte	−0.0116	–	0.0002	(	−0.0053	±	0.0026	)
AST	−0.0048	–	0.0106	(	0.0008	±	0.0026	)
ALT	−0.0044	–	0.0129	(	0.0032	±	0.0033	)
RBC	−1.6189	–	0.6399	(	−0.6277	±	0.4279	)
Band	0.0003	–	0.2040	(	0.0758	±	0.0364	)
Monocyte	−0.0715	–	−0.0263	(	−0.0491	±	0.0082	)
Eosinophil	−0.2581	–	−0.1520	(	−0.1999	±	0.0215	)
Basophil	−0.1549	–	0.5387	(	0.1284	±	0.1594	)
ANC	0.0731	–	0.2211	(	0.1424	±	0.0271	)
Segment+Band	−0.0282	–	−0.0152	(	−0.0212	±	0.0032	)

WBC, white blood cell; ANC, absolute neutrophil count; MCV, mean corpuscular volume; MCH, mean corpuscular hemoglobin; MCHC, mean corpuscular hemoglobin concentration; ALT, alanine transaminase; AST, aspartate transaminase; CRP, C-reactive protein.

**Table 3 diagnostics-10-00307-t003:** The odds ratio of each variable and its 95%CI after repeating univariate logistic regression 100 times.

Variable.	OR	OR (95% CI)					Number of *p*-Value < 0.05
AGE	1.0156	(	0.9632	–	1.0685	)	0
WBC	0.9212	(	0.8239	–	1.0184	)	6
CRP	1.0039	(	1.0008	–	1.0069	)	88
Hemoglobin	1.0243	(	0.4922	–	2.3101	)	0
MCV	0.6104	(	0.3573	–	1.0680	)	33
MCH	4.4778	(	0.8719	–	21.6080	)	36
MCHC	0.3528	(	0.1058	–	1.2778	)	26
Platelet	1.0004	(	0.9985	–	1.0024	)	0
Lymphocyte	0.9947	(	0.9819	–	1.0080	)	0
AST	1.0008	(	0.9933	–	1.0071	)	2
ALT	1.0032	(	0.9944	–	1.0112	)	3
RBC	0.5837	(	0.0609	–	4.5206	)	0
Band	1.0795	(	0.9836	–	1.1761	)	35
Monocyte	0.9522	(	0.9110	–	0.9920	)	82
Eosinophil	0.8190	(	0.6873	–	0.9530	)	96
Basophil	1.1514	(	0.6484	–	1.8965	)	0
ANC	1.1534	(	1.0014	–	1.3394	)	43
Segment+Band	0.9790	(	0.9615	–	0.9972	)	82

WBC, white blood cell; ANC, absolute neutrophil count; MCV, mean corpuscular volume; MCH, mean corpuscular hemoglobin; MCHC, mean corpuscular hemoglobin concentration; ALT, alanine transaminase; AST, aspartate transaminase; CRP, C-reactive protein.

**Table 4 diagnostics-10-00307-t004:** The maximum value, minimum value, mean value, and standard deviation of multivariate binary logistic regression coefficients of significant risk factors.

Variable	minimum	–	maximum	(	Mean	±	SD	)
CRP	0.0020	–	0.0088	(	0.0045	±	0.0010	)
MONOCYTE	−0.0689	–	−0.0177	(	−0.0434	±	0.0093	)
EOSINOPHIL	−0.2244	–	−0.0950	(	−0.1789	±	0.0263	)
Segment+Band	−0.0346	–	−0.0112	(	−0.0244	±	0.0044	)

CRP, C-reactive protein.

**Table 5 diagnostics-10-00307-t005:** The maximum value of each performance index in our machine learning model by LR.

Performance Index	Max Value
Sensitivity (recall)	0.92 (@cost = 12)
Specificity (TN rate)	0.96 (@cos = 7)
Positive likelihood ratio	1.14 (@cost = 9)
Negative likelihood ratio	1.25 (@cost = 11)
Positive predictive value	0.013(@cost = 8)
Negative predictive value	0.993 (@cost = 9)
AUC	0.768 (@cost = 10)

AUC, area under ROC curve; TN, true negative.

**Table 6 diagnostics-10-00307-t006:** Distribution of several variables in febrile children under the age of 3 years old with or without bacteremia.

Variable	Bacteremia	Non-Bacteremia	*p*-Value
Demographics	N = 81	N = 5033	
Age	0.81 ± 0.78	1.12 ± 0.73	<0.001
Gender = Male	43 (53.09%)	2812 (55.87%)	0.617
Gender = Female	38 (46.91%)	2221 (44.13%)	
Laboratory Values			
WBC (10^3^/μL)	12.76 ± 6.40	11.40 ± 5.98	0.590
Neutrophil (%)	45.63 ± 21.57	48.65 ± 18.75	0.151
Lymphocyte (%)	34.58 ± 17.99	37.43 ± 17.19	0.140
Band (%)	0.87 ± 1.88	0.39 ±1.64	0.026
Monocyte (%)	8.72 ± 6.38	8.91 ± 4.36	0.797
Eosinophil (%)	0.71 ± 1.09	1.07 ± 1.53	0.004
Basophil (%)	0.24 ± 0.40	0.27 ± 0.35	0.591
ANC (10^3^/μL)	6.98 ± 3.20	7.36 ± 2.82	0.228
Hemoglobin (g/dL)	11.20 ± 1.49	11.73 ± 1.31	<0.001
MCV (fL)	80.43 ± 8.84	79.07 ± 7.39	0.171
MCH (pg)	27.09 ± 3.25	26.63 ± 2.79	0.207
MCHC (g/L)	33.64 ± 1.05	33.64 ± 0.98	0.983
Platelet (10^3^/μL)	296.43 ± 124.97	286.64 ± 112.58	0.438
AST (U/L)	46.43 ± 39.59	42.80 ± 38.53	0.400
ALT (U/L)	28.43 ± 25.84	25.26 ± 30.02	0.345
CRP (mg/L)	60.24 ± 71.39	31.47 ± 48.95	0.001

WBC, white blood cell; ANC, absolute neutrophil count; MCV, mean corpuscular volume; MCH, mean corpuscular hemoglobin; MCHC, mean corpuscular hemoglobin concentration; ALT, alanine transaminase; AST, aspartate transaminase; CRP, C-reactive protein.

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
