# Peer review of "Using Machine Learning to Predict Bacteremia in Febrile Children Presented to the Emergency Department"

_diagnostics, 2020, doi:10.3390/diagnostics10050307_

Round 1

Reviewer 1 Report

The paper analyses the contribution of machine learning in predicting bacteremia in children in ER department. It shows that is possible to create an algorithm which learns from input data in order to make a prediction with clinical relevance. It is important from a practical point of view, since the recommended routine blood culture test is rarely positive.

The database is very large and analysis is clear and solid. But there are some important issues that need to be addressed and adjusted.

  1. Page 2, line 68. 0.86% febrile children had true bacteremia. Can you add some comments about this small sample size? How relevant is for the final analysis?
  2. Page 2, subchapter 2.2. Did you use a supervised or an unsupervised learning? Because the latter opens the way to discovering new phenotypes. Can you discuss the significance of deep learning, the most powerful machine learning algorithm? It learns from raw data input, as you do have a large cohort, and with useful patterns enable accurate task decision making.
  3. Page 2, line 84. Your predictive model is based on 10 standard laboratory features. Are there any other significant information available, such as clinical symptoms, location of infection? It might help your prediction model.
  4. Page 4, table 1 A. How do you explain the significantly lower percentage of neutrophils in children with bacteremia?
  5. Page 6. Line 139. Are there any data about vaccination? This might explain the 0.1% S. Pneumoniae positive bacteremia.
  6. Page 6, Discussion. Can you add some comments from the table 1 A and 1 B, regarding the significantly differed values of AND, platelet and eosinophils?
  7. Page 8, from line 208. Can you suggest what clinical information from medical history, available in a database, that can be data-mined in order to improve your predictive mode?
  8. Page 8, 260. The limitation of any machine learning is the dependence on the “training” data and the risk of making errors with future cases if the training cohort did not include similar case. Have you taught of testing the reproducibility of the task performance quality on an independent cohort?
  9. Page 9, conclusion. You suggest using your 5 group of risk model as a decision support tool. How do you think this can be applied in clinical practice? As a computer app? As a data visualization? Interpretable in a time-efficient manner? Enabling a decision-making process?

Author Response

The authors have made all-out effort in the revised manuscript and again would like to thank the reviewer for his/her kind review, which allowed us to considerably improve this work. We truly enjoy this submission and revision process and hope that these revisions will meet with your approval and that the revised manuscript will be considered for acceptance in the Diagnostics. We look forward to hearing from you soon.

Reviewer 2 Report

This study was aimed to investigate to build machine learning models to predict bacteremia in children with fever who visit the Pediatric Emergency Department and were subsequently diagnosed with blood-culture based methods. Since blood cultures in febrile children rarely identifies a causative pathogen, the authors performed a retrospective cohort study from January 2008– December 2015 that included infants younger than 18 years old with febrile.
The study included 16967 patients, of whom 146 had true bacteremia (as confirmed by blood culture). Age, percentage of neutrophil and band, ANC, Hb, platelet, and CRP were statistically different between bacteremia and non-bacteremia groups. Since most children with bacteremia were under age three a subgroup study was performed. Compared to those in the non-bacteremic group, the bacteremic group had a lower mean age; higher levels of C-reactive protein (CRP), higher band percentage and lower levels of hemoglobin and eosinophil percentage. Most common isolated pathogens were Salmonella entericae, Escherichia coli, and Group B Streptococcus. Using the predictive model, a risk rank was established (class 5 had the highest probability of having bacteremia while class 1 had no risk). Authors' final conclusion was that prediction models can help physicians determine whether patients have bacteremia.

 The article although interesting from the clinical point of view is not new as the authors themselves describe in the discussion. However, the high number of patients included provides great reliability to the results obtained. Therefore, in my opinion, the article could be eligible for publication as long as the authors answer the following question:

1) The authors are comparing two methods (machine learning vs. blood culture) for the diagnosis of bacteremia. It would be interesting to carry out (and show) a more complete statistical study by including univariate and multivariate binary logistic regression analyses to identify the significant risk factors for bacteremic, and the odds ratio (OR) and 95% confidence interval (CI). If variables are found to be significant in the univariate analyses then they must be included in the multivariate regression analysis. Receiver operating characteristic (ROC) curves must then be analyzed (and show again) to examine the predicting capacity of machine learning parameters and multivariate model in bacteremic which will allow establish optimal cutoff values.

By comparing these two methods sensitivity, specificity, positive and negative predictive values, positive and negative likelihood ratios must be analyzed which will provide the real value of the machine learning based method in bacteremia diagnosis.

2) One of the advantages of blood cultures is that they provide information about bacteria causing the infection. Would the machine learning method be useful to know which bacteria is responsible for the infection? Are there differences depending on the infecting bacteria?.

3) In the abstract line 27, the last sentence is confusing "... may reduce healthcare resources." I think that authors mean that may reduce unnecessary  expenses. Authors should explain that.

4) Line 85; LR and SVM appear for the first time in the text, so complete names should be included. 

Author Response

(The authors gave the same response as above.)

Round 2

Reviewer 1 Report

The new version of the paper provides sufficient improvement and managed to respond at all suggested ideas from my review!

Author Response

We thank the reviewer's approval of our revision.

Reviewer 2 Report

Dear Authors:

Thank you very much for reviewing the article according to the suggestions. However, despite having made an important statistical analysis this is not reflected in the manuscript. All tables, graphs, etc. should be shown in the results section of the manuscript, discussed in the discussion section, and appear in the abstract.

I would suggest the authors make a new table (comparing the new and old diagnostic factors) indicating precisely sensitivity, specificity, positive and negative predictive values, positive and negative likelihood ratios ... On the other hand, I miss a conclusion paragraph indicating whether and how the new diagnostic technique improves upon the one already available, so that a reader can draw appropriate conclusions.

Author Response

We would like to thank the reviewer 2 for his/her critiques and comments for our manuscript (Manuscript ID: diagnostics-776446). We have carefully followed the reviewer’s suggestions to revise the article with all-out effort and have addressed the changes that we have made item by item.

The authors again would like to thank the reviewer 2 for his/her kind review, which allowed us to considerably improve this work. We truly enjoy this submission and revision process and hope that these revisions will meet with your approval and that the revised manuscript will be considered for acceptance in Diagnostics. We look forward to hearing from you soon.

Round 3

Reviewer 2 Report

The authors have satisfactory answered to all the questions addressed.